# Beam Damage Assessment Using Natural Frequency Shift and Machine Learning

**DOI:** 10.3390/s22031118

**Published:** 2022-02-01

**Authors:** Nicoleta Gillich, Cristian Tufisi, Christian Sacarea, Catalin V. Rusu, Gilbert-Rainer Gillich, Zeno-Iosif Praisach, Mario Ardeljan

**Affiliations:** 1Department of Engineering Science, Babeș-Bolyai University, Str. M. Kogălniceanu 1, 400084 Cluj-Napoca, Romania; nicoleta.gillich@ubbcluj.ro (N.G.); cristian.tufisi@ubbcluj.ro (C.T.); zeno.praisach@ubbcluj.ro (Z.-I.P.); 2Department of Computer Science, Institute of German Studies, Babeș-Bolyai University, Str. M. Kogălniceanu 1, 400084 Cluj-Napoca, Romania; christian.sacarea@ubbcluj.ro (C.S.); vasile.rusu@ubbcluj.ro (C.V.R.); 3Doctoral School of Engineering, Babeș-Bolyai University, Str. M. Kogălniceanu 1, 400084 Cluj-Napoca, Romania; mario.ardeljan@ubbcluj.ro

**Keywords:** damage detection, linear regression, random forest, artificial neural network, training parameters, natural frequency

## Abstract

Damage detection based on modal parameter changes has become popular in the last few decades. Nowadays, there are robust and reliable mathematical relations available to predict natural frequency changes if damage parameters are known. Using these relations, it is possible to create databases containing a large variety of damage scenarios. Damage can be thus assessed by applying an inverse method. The problem is the complexity of the database, especially for structures with more cracks. In this paper, we propose two machine learning methods, namely the random forest (RF), and the artificial neural network (ANN), as search tools. The databases we developed contain damage scenarios for a prismatic cantilever beam with one crack and ideal and non-ideal boundary conditions. The crack assessment was made in two steps. First, a coarse damage location was found from the networks trained for scenarios comprising the whole beam. Afterwards, the assessment was made involving a particular network trained for the segment of the beam on which the crack was previously found. Using the two machine learning methods, we succeeded in estimating the crack location and severity with high accuracy for both simulation and laboratory experiments. Regarding the location of the crack, which was the main goal of the practitioners, the errors were less than 0.6%. Based on these achievements, we concluded that the damage assessment we propose, in conjunction with the machine learning methods, is robust and reliable.

## 1. Introduction

Nondestructive damage detection methods have received increasing attention in recent decades and have become a central research topic for scholars and practitioners belonging to the structural health monitoring community. The principles of vibration-based techniques, which are nowadays very popular, can be found in [1,2,3,4,5]. These methods are based on the deterministic relation between the damage characteristics (mainly location and severity), and changes in the modal parameters (natural frequencies, mode shapes, and curvatures). Among the modal parameters, the natural frequencies are the easiest to determine and require the involvement of relatively cheap and very robust instrumentation. Moreover, estimating the natural frequencies requires a limited number of sensors compared to methods based on mode shapes [6]. For this reason, in this paper, we focus on the analysis of natural frequencies to detect damage.

Temperature changes can affect the natural frequencies of structures. For accurate damage detection, it is important to remove the effects of temperature on the natural frequencies. An algorithm with this aim was presented in [7], where a successful damage assessment was performed for the case of changing temperatures. Mass changes also affect the natural frequencies of structures. In [8], an analytical approach was developed for a mass-spring-damper system that demonstrates how to localize mass change.

Detection and measurement of damage features for different crack types in slender beams using modal analysis was studied in [9]. Assessment of L and T-shaped cracks as well as delamination in bi-metal structures using natural frequencies was successfully performed in [10]. An assessment of corrosion and the analysis of the structural capacity of corroded I girders that belong to steel bridges was studied in [11].

In some real cases, multiple damages can occur at once, and the structures can have multiple supports. A complex study devoted to calculating the natural frequencies and mode shapes of multi-span beams is found in [12], and in [13], a method to assess cracks in continuous beams was presented. A study devoted to identifying multiple damages of multi-span bridges based on influence lines was presented in [14].

The complexity of damage (the crack shape and the orientation, the associated loss of mass, the number of cracks), as well as the complexity of the structures (multi-supported beams, skeletal structures, the use of nonisotropic materials), increase the dimension of the dataset to be processed for damage detection. However, for all cases mentioned above, we determined reliable relations to also consider the different configurations of the structure (multi-span beams or frames), the temperature effects, and the shape of the crack (L or T shape, delamination). This makes the damage detection methods able to take into consideration particular cases, which usually occur in real applications, but makes the recognition of patterns more difficult. For this reason, researchers who develop methods for detecting damages by employing vibration-based methods [15] are increasingly using artificial intelligence (AI) to analyze large amounts of data. AI has now reached a level where it can recognize the characteristics of objects as well as alterations of structures by considering different types of data and can automatically learn the needed criteria [16]. A study of the use of SHM methods is presented in [17,18] in which the authors presented an image processing model capable of monitoring microseismic events [17] or pavement monitoring [18] both of which achieved great strides in enabling the widespread use of artificial intelligence solutions, although many of these continue to face challenges related to large volumes of graphics data. Another approach is the use of vibration-based damage detection methods, which are global and do not require access to the damaged area.

The use of Random Forests (RF) and data fusion for structural damage detection are proposed in [19]. More recently, an RF model was used to predict the location of potential damage on asphalt pavement [20]. Here, RF data mining was used to analyze the interrelationships of variables. Another approach is found in [21]. The damage-sensitive features were extracted from raw sensor data using the cross-correlation function and wavelet packet decomposition. RF and other ensemble learning algorithms such as XGBoost are used to train the damage pattern classifier.

The artificial neural network (ANN) is a commonly applied technique for SHM. A method to identify the damage location and its severity in a ten-floor structure, employing an auto-associative neural network combined with transmissibility is proposed in [22]. The development of a method to detect damages in a truss structure using an ANN is the subject of the research presented in [23]. A method to predict crack width for thick, as well as for thin, concrete elements based on the feed-forward backpropagation and radial basis neural networks is proposed in [24]. Regarding the use of neural networks for condition monitoring, we mention a technique involving a new convolutional neural network (NCNN) that was proposed and successfully applied to detect bearing faults [25]. 

Despite successful assessment of damage, it is worth mentioning that the damage considered in most studies presented in the literature either manifests on an element with significant extent of damage along the beam [22] or has significant severity: 22% to 68% in [26] and 20% to 80% in [27]. For skeletal structures, the damage manifests on one or more structural elements, see, for example, [19,20,21,23]. 

In prior research, we deduced a relation to calculate the natural frequencies of beams with known crack parameters. This relation is applicable also for beams with non-ideal boundary conditions. Being an analytical relation, it permits creating, easily and rapidly, a database with patterns for a multitude of damage scenarios. The database contains the relative frequency shifts (RFS) for eight out-of-plain vibration modes for all given damage locations, severities, and fixing conditions. It was used to train RF and ANN with a huge amount of data. The training was first performed for scenarios covering all locations along the beam and afterwards for locations on a specific segment. The proposed damage detection methodology presumes an initial/coarse assessment to find the crack location, followed by a second/fine assessment targeting the accurate crack location and severity.

To evaluate the effectiveness of the proposed approach, two examples, comprising numerical simulations and laboratory experiments on steel beams were performed. The results obtained by involving RF and ANN were compared; both methods led to the correct location and quantification of the damage, regardless of the changes in the fixing conditions. It was possible to detect cracks with a much smaller depth than those reported in the literature because a large amount of the calculated, and consequently noiseless, data was used for training. To the best of our knowledge, there are no artificial intelligence-supported crack detection methods that apply to beams with non-ideal boundary conditions or that can detect the weakening of the clamping. 

The paper is organized as follows. After an introductory section, we present the theoretical background which permits the development of a database consisting of an INPUT and a TARGET part. The methodology of RF and ANN, along with the training process based on the database are introduced in Section 3. In the next two sections, we prove the efficiency of the proposed approach. In Section 4 we describe the numerical simulations performed on a steel cantilever beam, done to find the natural frequency shifts due to several damage cases. Then, we apply the two machine learning methods. In Section 5 we repeat the approach for laboratory-scale beams. Eventually, conclusions are formulated.

## 2. Creating the Database for Machine Learning (ML)

As a crack propagates in a structure, it produces an alteration of structural stiffness. This alteration produces a change in modal parameters, of which the most obvious and easy to measure are changes in natural frequencies. There is a deterministic relationship between stiffness change and the decrease in the natural frequencies of the structure; therefore, the position and depth of the crack can be identified if the changes in the frequencies produced by the defect are known. We proposed a robust and easy-to-use mathematical relation to predicting the natural frequency $f_{i-D}(x,a)$ of a cracked beam [28]. This relation relies on the features of the healthy beam, namely, the natural frequencies $f_{i-U}$ and the modal curvatures $\phi_{i}^{\prime \prime }(x)$ of the out-of-plain vibration modes, and the damage severity γ(a), respectively. The mathematical relation, following [28], is
(1)$$f_{i-D}(x,a)=f_{i-U}\left\{1-\gamma (a){\left[\bar{\phi_{i}^{\prime \prime }}(x)\right]}^{2}\right\}.$$

In this relation, we denoted the crack position with *x*, the crack depth with *a*, and *i* stays for the mode number. From Equation (1) we can deduce the relative frequency shift (RFS), which is the normalized frequency drop due to a crack, as
(2)$$\Delta {\bar{f}}_{i-D}(x,a)=\frac{f_{i-U}-f_{i-D}(x,a)}{f_{i-U}}=\gamma (a){\left[\bar{\phi_{i}^{\prime \prime }}(x)\right]}^{2}.$$

We find the two terms in the right part of Equation (2) as follows. The crack severity is calculated involving an energy method, which says that a beam with a crack will be able to store less energy than a similar healthy beam and will suffer a greater deformation if subjected to a force [29]. So, we can calculate the severity involving the mathematical relation proposed in [30], that is
(3)$$\gamma (a)=\frac{\sqrt{\delta_{D}(a)}-\sqrt{\delta_{U}}}{\sqrt{\delta_{D}(a)}}.$$

In this relation, we denoted $\delta_{U}$ the deflection of the healthy beam, and with $\delta_{D}(a)$ the defection of the beam with a crack that has depth *a.* Note that the severity is calculated for the crack located at the position where the biggest curvature (or bending moment, which is proportional to the curvature) is achieved. In consequence, the position of the damage for which the severity is calculated differs depending on the boundary conditions of the beam; for example, for the cantilever beam that is the subject of the study presented hereinafter this position is the fixed end. On the other hand, the severity depends solely on the crack depth *a*, so that for a given depth it is the same regardless of the boundary conditions. The relationship between the crack depth and the severity can be found before starting the damage detection procedure. When applying the damage detection method, we determine the damage severity and afterward determine the damage depth from the depth-severity relationship known a priori. So, knowing the deflection of the monitored structure is not necessary.

The effect of the crack position is controlled by the local curvature (or bending moment). This happens because the stress present in the affected slice is proportional to the bending moment. In consequence, a smaller effect on the frequency drop is obtained if the crack is not located at the particular position where the biggest bending moment is present. There are even situations in which the crack does not cause a decrease in frequency, namely when it is located at an inflection point [31]. To consider the effect of the crack position *x* on the frequency drop, we use the normalized squared curvature. By normalization, we assign value one to the curvature or bending moment for the location where the beam is the most requested and a subunit value for the rest of the positions. For the cantilever beam, the normalized curvature is
(4)$$\bar{\phi_{i}^{\prime \prime }}(x)=0.5\left\{\cos\left(\lambda_{i}x\right)+\cosh\left(\lambda_{i}x\right)-\frac{\cos\lambda_{i}+\cosh\lambda_{i}}{\sin\lambda_{i}+\sinh\lambda_{i}}\left[\sin\left(\lambda_{i}x\right)+\sinh\left(\lambda_{i}x\right)\right]\right\}.$$
where we denoted with $\lambda_{i}$ the eigenvalue of the *i*-th vibration mode. These values are indicated in the literature for different boundary conditions of the beam. Note that, other boundary conditions also lead to other curvature functions. 

Taking into account the above, we can summarize that:

The severity γ(a) depends on the crack depth *a* and is independent of the crack position *x*, boundary conditions, and vibration mode number *i*. Therefore, the severity for a beam with a given crack once calculated using Equation (3) is valid for that beam irrespective of the boundary conditions. In practice, we calculate the severity using data obtained from static finite element analysis for a cantilever beam because it presents an important deflection at the free end. A comprehensive description of the procedure to determine the correct severity is given in [32];The value of the normalized modal curvature $\bar{\phi_{i}^{\prime \prime }}(x)$ at the position *x* where the crack is located reduces the effect of the severity since at that position less stress is stored in the beam. This term depends on the vibration mode number *i* and the boundary conditions. Therefore, Equation (2) has a large degree of generality; it can be properly used for any support type if the correct curvature function $\bar{\phi_{i}^{\prime \prime }}(x)$ is employed. Hereinafter we exemplify the case of a cantilever, thus Equation (4) is used.

In Figure 1 we represent the RFS calculated using Equation (2) for vibration modes 1 and 3, for two crack depths. The RFS values for the fixed (left) end reflect the severity of the cracks when the cross-section is reduced by 12 and 20%, respectively. For the other locations, the effect of the crack is reduced in concordance to the local curvature. Because the normalized curvature is the unit for the crack located at the fixed end, here the relative frequency shift is equal with the crack severity and is not dependent on the vibration mode number. It is obvious that the bigger the crack depth, the bigger the severity and in consequence the RFS.

The above approach is valid for perfect boundary conditions. If the fixing is non-ideal, the frequency drop is bigger. In prior research, we demonstrated that the superposition principle is valid [33]. For a beam with two cracks, which have position $x_{1}$ and $x_{2}$, and depths $a_{1}$ and $a_{2}$, respectively, the frequency drop is calculated as the sum of the two RFS, that is
(5)$$\Delta {\bar{f}}_{i-D}(x_{1},a_{1},x_{2},a_{2})=\gamma_{1}(a_{1}){\left[\bar{\phi_{i}^{\prime \prime }}(x_{1})\right]}^{2}+\gamma_{2}(a_{2}){\left[\bar{\phi_{i}^{\prime \prime }}(x_{2})\right]}^{2}.$$

We simulate the non-ideal boundary condition, namely the weak fixing, as a crack at the clamped end. So, an additional rotation is possible [34], which leads to a frequency decrease. Knowing that the normalized modal curvature at the fixed end of the cantilever is $\bar{\phi_{i}^{\prime \prime }}(0)=1$, Equation (5) becomes
(6)$$\Delta {\bar{f}}_{i-D}(0,a_{1},x_{2},a_{2})=\gamma_{1}(a_{1})+\gamma_{2}(a_{2}){\left[\bar{\phi_{i}^{\prime \prime }}(x_{2})\right]}^{2}.$$

Hence, we can plot the RFS for the beam with non-ideal boundary conditions as shown in Figure 2. These are calculated using Equation (6), again for the vibration modes 1 and 3, and the same two crack depths (cross-section is reduced with 12% and 20%, respectively). In this figure, we can observe that in addition to the RFS caused by the crack we have the RFS caused by the non-ideal clamping. The two RFS, caused by crack and by weak clamping, are superimposed and the contribution of the weak clamping is marked with a red dashed line. 

The inverse method we developed to detect cracks implies, as a first step, calculating the RFS for numerous damage scenarios and creating a database. The database thus contains two distinct sections:TARGET elements, which are the local value of the curvature for a given position, the severity of the defect, and the severity corresponding to the weak fixing. These are placed on columns, the number of columns *m* corresponding to the desired number of scenarios. The index of the column is denoted *k*, thus *k* = 1,…,*m*;INPUT elements, which are the relative frequency shifts ${\mathrm{RFS}}_{i-k}^{c}$ calculated with the mathematical relations (2) or (6) for a chosen number of vibration modes *n*. These are also arranged in columns, each INPUT column corresponding to a TARGET column.

The second step is monitoring the structure, which presumes to measure the natural frequencies. If changes are observed, the frequency shifts calculated for the measured frequencies (${\mathrm{RFS}}_{i}^{m}$) are compared with the elements of the INPUT section of the database, i.e., ${\mathrm{RFS}}_{i-k}^{c}$. The column number *k* for which the best fit is obtained between the calculated and measured RFS indicates the crack position and severity. In this study, we propose a machine learning approach to find the best fit. The chosen methods are the random forest and the neural network.

## 3. Machine Learning Methods

In the last decade, artificial intelligence (AI) has become a frequently used term for applications that perform complex tasks that once required human intervention, such as structural health monitoring methods. In this section, we present the development and testing of two machine algorithms used for damage detection. The training data are generated using the method described in the previous chapter. The database contains 36,573 damage scenarios (number of columns in the INPUT and TARGET files). The number of rows in the INPUT file is eight, comprising the first eight out-of-plain bending vibration modes. The TARGET file consists of three elements, i.e., three rows, which are: the damage location, the damage severity, and the severity of the damage simulating the weak clamping. For this study, we considered 100 locations of the crack evenly distributed along the beam, the cracks having 17 levels of depth. The beam has both ideal and non-ideal clamping, for the latter situation seven cases of weak fixing are generated. It resulted in 36,573 damage scenarios. This number of data was selected after testing the influence of the dataset dimension on the accuracy of the predictions. We observed that larger databases led to better results. The two files of the database are presented in [35].

Because of the complexity of the application and the accuracy required for determining the exact position and severity of transverse cracks we propose a two-step approach. This means that after a coarse localization of the crack using a network trained for all damage scenarios, we apply a second check, this time for a model trained for the specific section of the beam on which the crack is found at the first step. As an example, if the crack is found involving one of the developed methods at 175 mm from the fixed end, we make the second check for a segment extended between 100 and 250 mm. For this segment, we train again the model for a limited number of inputs, considering just those which contain the crack position between the limits 100 and 250 mm. This facilitates obtaining a better model of the structure and, in consequence, more precise localization of the crack. 

To have a fast estimation, we divide the beam into nine segments, partially overlapped, and train the ML models a priori for these segments. Overlapping is used to avoid uncertainty regarding the choice of the right segment. The segments used, and the name of the network, are presented in Table 1.

The two ML methods and the settings applied for the training are presented in the next two sub-sections. For both methods, we use 70% of the data for training, 15% for testing, and 15% for validation. 

### 3.1. Random Forest

Decision trees have proven to be successful at exploring non-linear relationships between input and target variables [36]. Such trees work by splitting the dataset in instances that have a minimum amount of node impurity or are, in other words, homogenous. Purity here means that each leaf node represents data points that are in the same class and is defined as the sum of square deviations in class predictions. The biggest drawback of decision trees is that they can easily overfit. This can be mitigated by aggregating such trees to reduce variance.

RF is a technique that employs an ensemble of decision trees and can be used for regression and classifications tasks. The prediction made by an RF aggregates the output of individual trees into a single variable [37]. When building trees, the algorithm randomly selects a given number of features. This essentially prevents multiple decision trees that rely on the same feature. The process is repeated until a group of regression trees, each trained on a randomly selected subset of data, is created. This induced randomness is what compensates for the weakness of each tree.

The performance of an RF model can be tweaked by tuning a few key parameters, with some studies reporting that there is a significant benefit to tuning RF parameters away from their default settings [38,39]. Table 2 summarizes the most common hyperparameters of the RF model.

The number of trees in the forest should, in principle, be as high as possible but in practice, performance plateaus appear after a few hundred trees. In general, increasing the number of features considered in splitting a node will improve performance as each node will have now have a higher number of options to consider; a lower value will increase the chance of selecting features with small effects which in turn could lead to improved performance in cases where such feature would be masked. Parameter min_samples_leaf represents the minimum size of terminal nodes. A higher number will lead to small trees while a smaller leaf size makes the tree more prone to noise in data. 

The hyperparameters of the RF were tuned using randomized search with five-fold cross-validation (RandomizedSearchCV in Scikit-Learn), that chooses one of the possible values for each one of the hyperparameters and scores the estimator. The best estimator is used in the model. Table 2 also shows the tuned values for each parameter The training results using RF are presented in Figure 3 for the entire beam and sector 8, respectively.

### 3.2. Artificial Neural Networks

Numerous studies have shown the versatility and power of ANNs when applied to different computational tasks such as prediction or classification in many real-world applications [40]. In particular, they are universal approximators capable of detecting nonlinearities in an n-dimensional input. This is achieved by including a large number of nonlinear transformations between the input to an output mapping. A typical neural network consists of several connected neurons, organized in layers, as shown in Figure 4. A neuron will generally process information coming from its connections using a nonlinear activation function. A neural network is trained to perform a specific function by adjusting the values of the connections between its neurons.

Feedforward neural networks are ANNs where the topology is organized such that every neuron in a layer projects only onto subsequent layers. This topology excludes thus recurrent connections and essentially means that information flows through the network from one layer to another until it reaches the output.

By using the methodology presented in Section 2, we calculated the required data for training. The calculated data involves the INPUT data as the calculated RFS values for the first 8 transverse vibration modes and the TARGET data consisting of 3 values, i.e., crack position, crack severity, and the severity for the weak clamping. 

The ANN is developed using Matlab software, as shown in Figure 5. A feedforward-backpropagation network type is employed, for which we choose: the *Bayesian regularization* training function, the *Levenberg–Marquardt* learning function, the *mean squared error (MSE)* performance function, and the *hyperbolic tangent sigmoid* transfer function. This combination of Bayesian regularization and Levenberg–Marquardt training leads to regularisation and essentially makes the network difficult to overfit [41].

All networks used in this study consist of one input layer with 8 neurons, three hidden layers with 30 neurons each, and an output layer with 3 neurons. An example is shown in Figure 6.

The obtained training performance for *network 1* is presented in Figure 7 in comparison with one of the particular networks, Sector 9 shown in Figure 8.

### 3.3. Evaluation of the Models

In this study, two criteria were used for evaluating the accuracy of the models relative to the ones obtained through FEM analysis and experimental tests, *error*(*x*) for the position and *error*(*γ*) for the severity. The error in the position is
(7)$$error(x)=\frac{x(r)-x(e)}{1000}\cdot 100\%$$

Here, *x*(*r*) is the real position of the crack expressed in mm, *x*(*e*) is the estimated position using the ML models and 1000 represents the total length of the beam in mm. 

The severity error is calculated as
(8)$$error(\gamma )=\frac{\gamma_{2}(r)-\gamma_{2}(e)}{1}\cdot 100\%$$

In this relation, *γ*(*r*) is the real severity of the crack, *γ*(*e*) is the estimated severity of the crack found using the ML models, and 1 represents the maximum value that can be achieved by the crack severity.

In addition to the two above-mentioned errors, we evaluate the capacity of the models to detect weak clamping. The possible responses are *false* or *true*.

## 4. Numerical Validation

For testing the reliability of the damage detection methods described in the previous section, we first involve the finite element method (FEM). Modal analysis is performed using the ANSYS software. We generate a prismatic steel cantilever beam, like that shown in Figure 9, with dimensions: length *L* = 1000 mm, width *B* = 50 mm, and thickness *H* = 5 mm. The assigned material is S355 JR steel with a modulus of elasticity *E* = 2 × 10^5^ MPa and a density of 7850 kg/m^3^.

The target was to find the first eight natural frequencies for the out-of-plain vibration modes, for the healthy beam and the beam with different damages, respectively. Both ideal clamping and non-ideal clamping are simulated. 

To implement the crack and simulate the effect of the weak clamping, we involve separate elements which are not fixed on one surface, as shown in Figure 10. In this way, we assign the same mass for the beams with damage and non-ideal clamping and yet obtain discontinuities. For simulating real-life scenarios where in most cases measurement data can contain noise, we have also considered different meshing sizes for the damage scenarios to have small differences in the frequency results. The maximum edge size is set to 2 mm for scenarios 1–5 and 1 mm for scenarios 6–10.

The FEM model is composed of the beam body and the two parameterized elements, which have a thickness equal to the crack depth and the length of 6 mm. The elements are used to create discontinuities by setting the free condition for the faces where the crack is intended. 

The frequencies obtained from the FEM analysis for the undamaged and damaged beam cases are presented in Table 3. The complete dataset is presented in [42].

The variables used for the studied damage scenarios are the crack location, which is measured from the fixed end, and the proportion in which the clamped end is affected. The weak clamping is replicated by adding the separate element of known thickness ${\bar{a}}_{1}$ to the fixed end. Thus, ${\bar{a}}_{1}=a_{1}/H$ (%) replicates the weak clamping. The RFS values obtained for the studied damage scenarios are presented in Table 4, where crack position *x*, crack depth $a_{2}$ and weak clamping ${\bar{a}}_{1}$ represent the output values.

The outputs obtained using the two ML methods based on the RFS values (Table 4) are presented in Table 5, Table 6, Table 7 and Table 8. Table 5 contains the results from the preliminary ML *random forest* model, and Table 6 the results obtained using the refined *random forest* model. In Table 7 we present the results obtained involving coarse estimation with *network 1*, and in Table 8 are the results obtained with the specific ANN networks chosen for the particular cases after the coarse estimation. 

The severity of the cracks with depth *a* = 1 mm, *a* = 1.2 mm, and *a* = 0.6 mm is: *γ*_2_(1) = 0.0033459, *γ*_2_(1.2) = 0.0051239 and *γ*_2_(0.6) = 0.0011911.

One can observe that, after the first check, the largest error in estimating the crack position is 7.51%, which is unacceptable. The severity is precisely found, the errors being less than 1.17%. The method was not able to indicate correctly if the clamping is non-ideal. In consequence, this machine learning method is not qualified to assess the crack position and severity at this stage.

Localization after the second check with the RF method is accurate, errors being below 1%. Estimating the severity of the defect is done also with small errors, up to 0.88%. As shown in Table 6, weak clamping is detected incorrectly for some damage scenarios.

Assessing the crack with the ANN method is also made in two steps. First, a coarse estimation is made with *network 1,* which considers the locations along the entire beam. Here, we coarsely estimate the damage location and choose the segment on which the crack is supposed to exist and perform a second check for the network trained for scenarios available on this segment. 

After the first check, the largest error obtained is 3.3% for the position, and small errors were obtained for the crack severity. The model wrongly detects non-ideal clamping for the first five damage scenarios. But, at this stage, we can estimate the segment on which the crack occurred with high confidence.

Knowing the approximate position of the crack we can select the appropriate segment and use the network trained for this particular segment. The results obtained in the second step are given in Table 8. One can observe that now the largest error obtained is 0.5% for the position and 0.07% for the crack severity. In addition, the model can detect non-ideal clamping with high accuracy. These results allow us to conclude that the two-step ANN method is more efficient in crack assessment.

## 5. Experimental Validation

To validate the developed ANN models, experimental studies were carried out by measuring the first natural frequencies for five steel beams with the dimensions indicated in the previous section. In the first five damage scenarios, the beams were perfectly clamped. Furthermore, the capability of the developed method to detect non-ideal clamping was also tested for the damaged beams by mounting rubber blocks between the jaws of the vise and the beam (Figure 11). The transverse cracks are produced by wire-cut electrical discharge machining, a method chosen to obtain cracks width of 0.08 mm.

The acceleration signals are recorded with a Kistler 8772 accelerometer mounted on the tested beam (5). It transmits the measured signal through the analog-to-digital conversion module NI9234 mounted in a compact chassis NIcDAQ-9175 (3) which is connected to laptop (2). The beam (5) is fixed into the vise (7) and to excite it at the desired resonant frequencies we used the method described in [43]. An audio excitation system consisting of a loudspeaker (6), amplifier (5), and dedicated software installed on a second laptop are used (1). The experimental setup is shown in Figure 12.

The data is processed using the LabVIEW software installed on laptop (2). We employ a VI designed to acquire the acceleration signal, plotted in the upper graph in Figure 13. The coarse natural frequencies are estimated; in the bottom graph in Figure 13 we present the spectrum of the signal acquired with the excitation set to produce resonance for the 4th out-of-plain vibration mode, at around 140 Hz.

The natural frequencies are estimated one by one, with high accuracy, after a procedure described in [44]. This is imperious necessary for detecting the small changes induced by the crack. Following the procedure, the acquired signal is imported in a Python application, named PyFEST, developed by our research team that uses a method which truncates the acquired signal and calculates the DFT values for different frequency resolutions, and the results are graphically shown in Figure 14. The Python code to estimate the frequencies is given in [45].

### 5.1. Perfect Clamping Experiments

In the first experimental study, each beam is mounted into a vise, thus achieving a rigid clamping. The resulted frequencies for the undamaged beams are shown in Table 9. 

Several damage scenarios are generated by saw cutting each of the five beams at different positions and depths thus replicating a transversal crack. The position and depth for every scenario are according to Table 10.

By using the measured frequencies for the beams in an undamaged and damaged state, we obtain the RFS values with Equation (2); the results are presented in Table 11.

The RFS data is fed to the coarse ML models to obtain the estimated crack position and severity for the five damage scenarios. The results obtained after the first estimation are shown in Table 12 for the RF model and in Table 13 for the ANN model.

We identify the segments in which cracks were found and select the corresponding networks. Afterward, by employing the enhanced ML models we obtain more precise results, as presented in Table 14 for the refined random forest model and in Table 15 for the enhanced ANN model.

One can observe from Table 14 and Table 15 that, for the case of ideal clamping, the damage location and severity are found with high accuracy irrespective of the employed method. Remarkably, ideal clamping is detected in all cases. 

### 5.2. Improper Clamping Experiments

This study concerns the ability of the proposed approach to detect cracks in two different cases:

The structure has initially ideal boundary conditions, but after a while, a crack occurs and an alteration of the fixing system is present, i.e., the clamping becomes non-ideal;The structure has from the beginning non-ideal boundary conditions, so the fixing system remains unchanged, and is affected after a while by a crack.

The experiment is made for test beam 1. Improper clamping was ensured by mounting rubber blocks between the jaws of the vise and the test beam, as shown in Figure 10. The crack is generated at distance $x_{2}=98$ mm and has depth $a_{2}=2.5$ mm, The natural frequencies were obtained from the measured data. The results for the undamaged beam with ideal and non-ideal clamping, as well as the frequencies for the test beam 1 with non-ideal fixing and a crack are shown in Table 16.

The calculated RFS values are presented in Table 17, first for the undamaged beam with ideal fixing (case 1i) and afterward, for the undamaged beam with initially non-ideal fixing (case 1n-i), In both cases, the damaged beam has non-ideal clamping. 

The crack assessment is made employing the two ML methods, in compliance with the two evaluation stages. The results for *case 1i* are presented in Table 18, while the results for *case 1n-i* are presented in Table 19.

Reviewing Table 18 and Table 19, we can conclude that for the beam with changing boundary conditions, the crack position and severity were found accurately after the second step. The best result is obtained using the ANN, which located the crack with an error of 0.11% and found the severity with an error below 0.1%.

The changed boundary conditions are detected with both ML methods. Dissimilarly, the case when the beam initially had a weak fixing was properly solved just by the ANN method. The errors accuracy, in this case, is sustained by errors less than 0.16%, for position and severity as well. Both ML methods did not find an alteration of the fixing in the vise, thus we consider the response to not be detected as true.

## 6. Conclusions

In this paper, we applied RF and ANN to identify the crack location and severity in a steel cantilever beam with a rectangular cross-section. The beam had different levels of fixing, including ideal and non-ideal clamping, and we also aimed to find out if the fixing condition changed. To train the network we used as inputs the RFS found, applying an original method, as the crack parameters and the fixing condition, respectively. To improve the accuracy of the assessment method, we performed the crack assessment in two steps: (i) in the first step we applied coarse estimation for the network trained for all damage scenarios to identify the region in which the crack occurred; (ii) in the second step we applied a fine estimation, for the network trained for a specific segment of the beam which included the damaged region to determine the crack position and severity, along with the changes in clamping, if any.

Both ML methods were able to learn and classify new data with characteristics comparable with that of the training data. After training, the networks were successfully used to assess other scenarios, which came from FEM simulation and laboratory experiments. We found that, for all cases, the ability to accurately detect the crack location and severity increases when using the two-step assessment. After the second step, the errors in locating the crack were below 0.11% for the ANN method and 0.53% for RF, and the errors in assessing the crack severity were less than 0.08% for the ANN method and 1.28% for RF. Thus, the ANN led to better results. The smallest crack we found had a 0.6 mm depth in a beam with a thickness of 5 mm. We did not search for smaller crack depths because of the training data we produced, and the knowledge that the ANN method is more efficient in interpolation than extrapolation. 

As a limitation, we mention that the proposed method is based on mathematical relations to calculate the frequency variation due to cracking. Thus, the method is easy to apply for relatively simple structures, such as beams. For more complex structures it is necessary to determine the RFS using FEM analysis or from experiments, which can be a laborious activity and provide results possibly affected by noise.

Further investigation should be conducted to evaluate the ability of the proposed approach to identify cracks in the early stage and on large-scale structures.

## Figures and Tables

**Figure 1 sensors-22-01118-f001:**
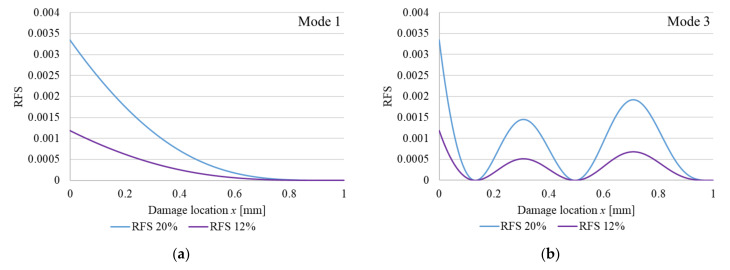
Plotted RFS functions for out-of-plain bending vibrations of a cantilever beam with perfect fixing: (**a**) vibration; mode 1; (**b**) vibration mode 3. The functions are plotted for a cross-section reduction of 12% and 20%, respectively.

**Figure 2 sensors-22-01118-f002:**
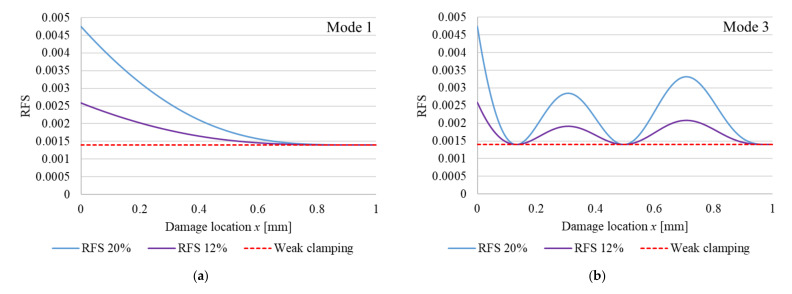
Plotted RFS functions for out-of-plain bending vibrations for a cantilever beam with non-ideal fixing: (**a**) vibration mode 1; (**b**) vibration mode 3. The functions are plotted for a cross-section reduction of 12%, respective 20%.

**Figure 3 sensors-22-01118-f003:**
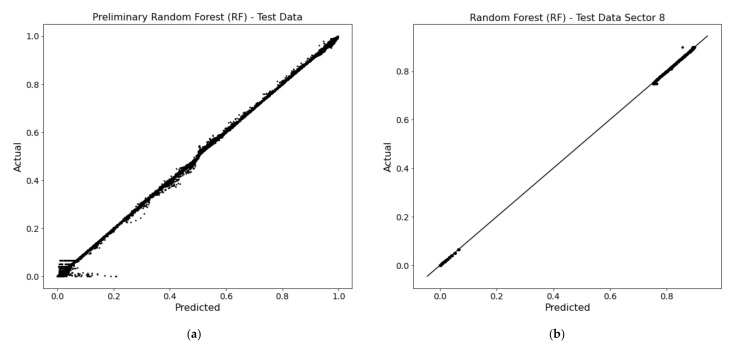
Random forest training results; (**a**) obtained for coarse estimation involving all input data; (**b**) obtained for accurate localization involving the *Sector 8* input data.

**Figure 4 sensors-22-01118-f004:**
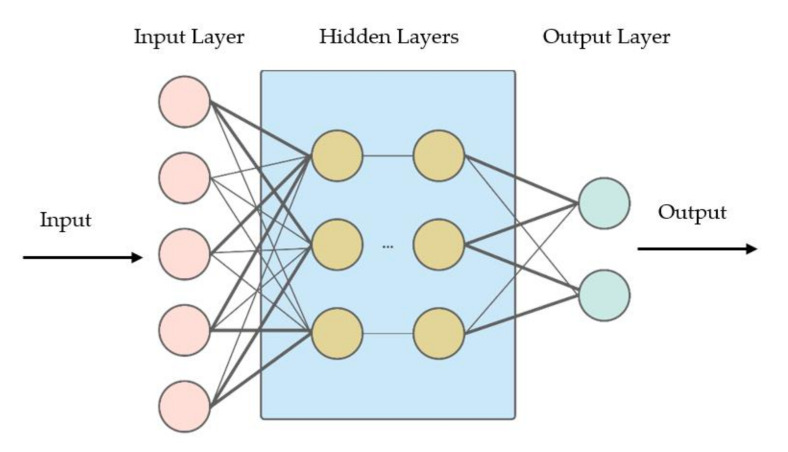
Typical neural network structure.

**Figure 5 sensors-22-01118-f005:**
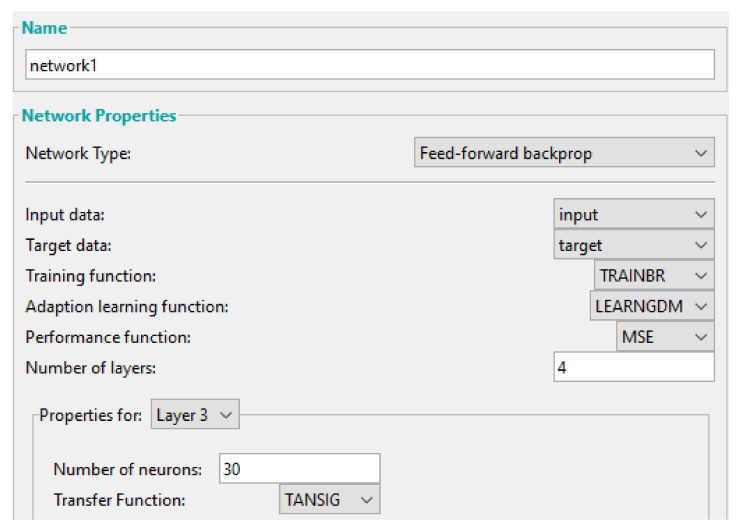
The setup for the training of network 1.

**Figure 6 sensors-22-01118-f006:**
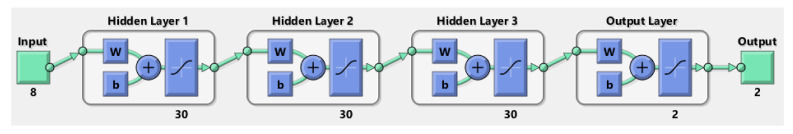
Network 1 configuration.

**Figure 7 sensors-22-01118-f007:**
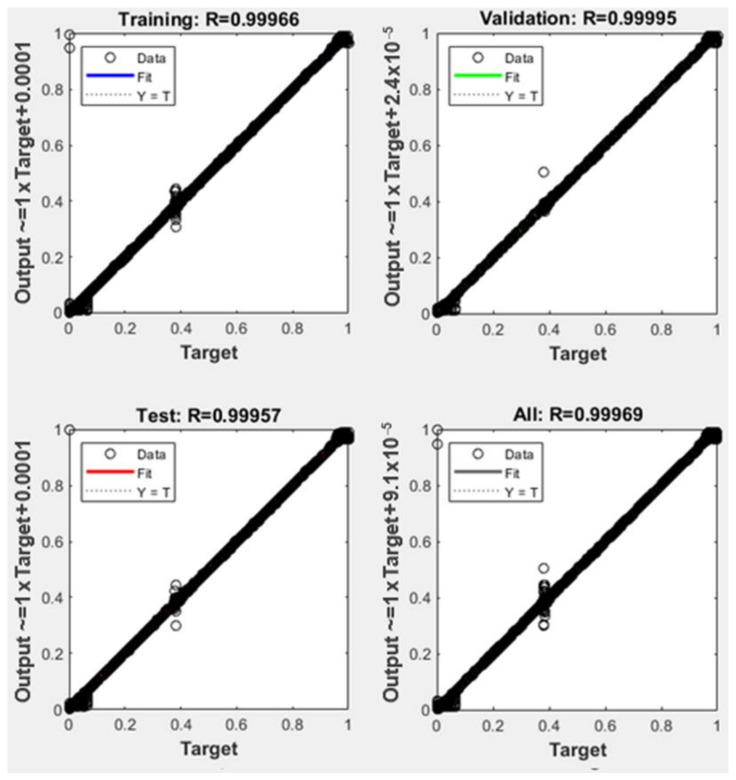
Neural network training results obtained for coarse estimation involving *network 1*.

**Figure 8 sensors-22-01118-f008:**
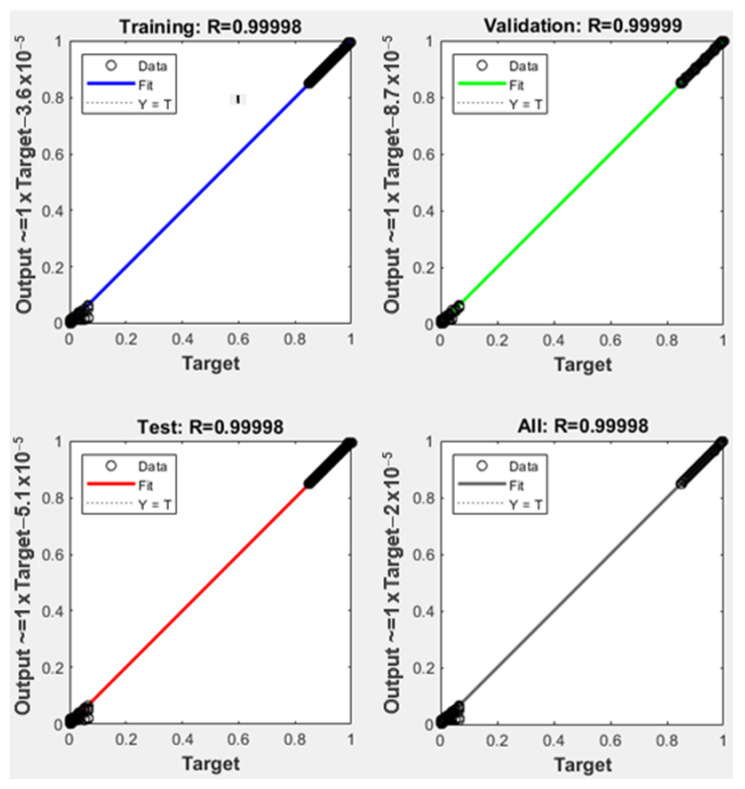
Neural network training results obtained for accurate localization involving the *Sector 9* network.

**Figure 9 sensors-22-01118-f009:**
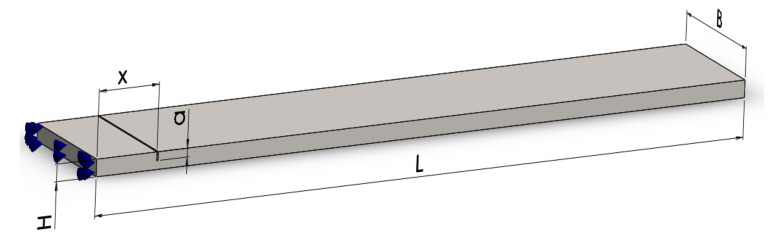
Main dimensions of the considered beam geometry with a transversal crack.

**Figure 10 sensors-22-01118-f010:**
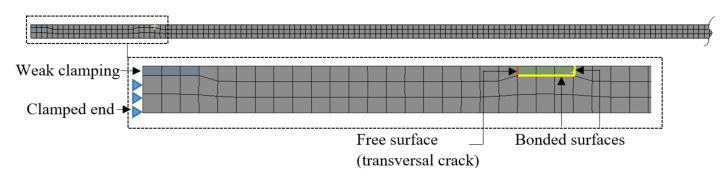
The finite element model of the damaged beam with weak clamping.

**Figure 11 sensors-22-01118-f011:**
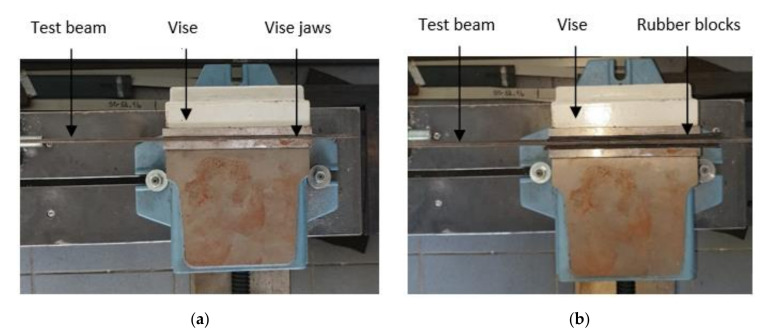
Clamping system: (**a**) rigid clamping obtained by direct fixing in the vise; (**b**) weak clamping obtained by intercalating rubber layers.

**Figure 12 sensors-22-01118-f012:**
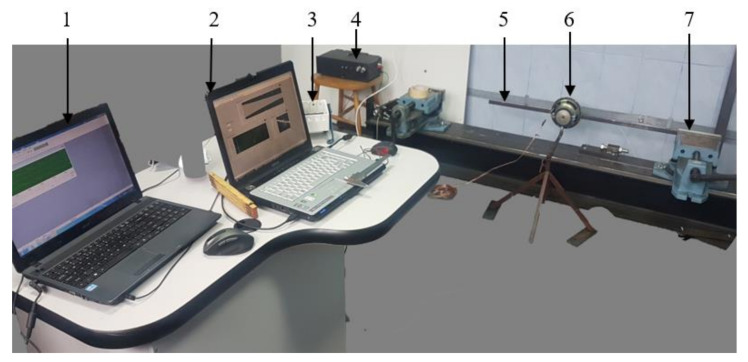
Experimental setup.

**Figure 13 sensors-22-01118-f013:**
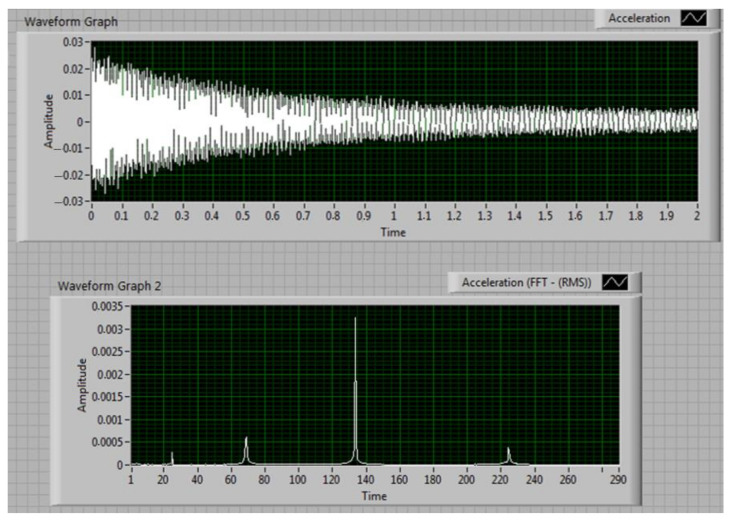
Mode no. 4 extracted signal for test beam 1.

**Figure 14 sensors-22-01118-f014:**
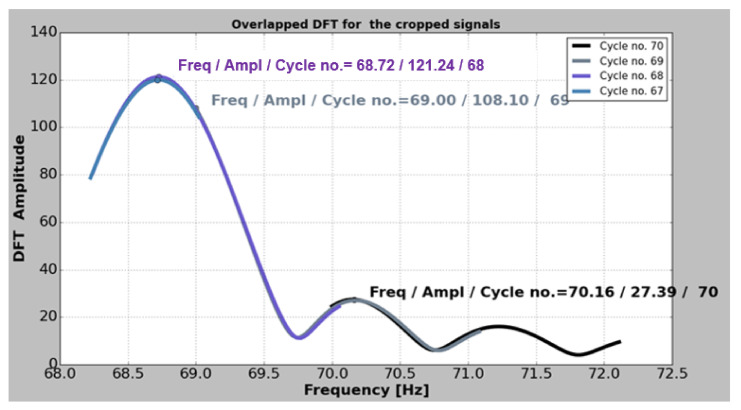
A window displaying the overlapped Discrete Fourier Transform (DFT).

**Table 1 sensors-22-01118-t001:** Segments used to train the machine learning models for enhanced damage detection.

**Segment limits**	0–150	100–300	250–400	350–500	450–600	550–700	650–800	750–900	850–1000
**Network name**	Sector 1	Sector 2	Sector 3	Sector 4	Sector 5	Sector 6	Sector 7	Sector 8	Sector 9

**Table 2 sensors-22-01118-t002:** RF hyperparameters.

Parameter	Meaning	Value	Range
n_estimations	The number of estimators in the forest	400	200–2000, step 200
max_features	max number of features considered for splitting a node	sqrt	-
max_depth	max number of levels in each decision tree	None	None; 10–110, step 10
min_samples_split	min number of data points placed in a node before it is split	2	2, 5, 10
min_samples_leaf	min number of data points allowed in a leaf node	1	1, 2, 4
bootstrap	method for sampling data points. True= bootstrap samples	true	true, false

**Table 3 sensors-22-01118-t003:** Obtained natural frequencies for the defined damage scenarios.

Scen.	Crack Pos. *x* (mm)	Crack Depth *a*_2_ (mm)	Weak Clamp. ${\bar{a}}_{1}\;(\%)$	Natural Frequencies Obtained for the First 8 Modes							
				Mode 1	Mode 2	Mode 3	Mode 4	Mode 5	Mode 6	Mode 7	Mode 8
0	Undamaged			4.09	25.627	71.757	140.63	232.53	347.46	485.47	646.59
1	100	1	0	4.081	25.606	71.745	140.631	232.472	347.229	484.947	645.751
2	150	1	0	4.082	25.620	71.754	140.552	232.257	347.024	485.083	646.448
3	400	1	0	4.087	25.601	71.708	140.588	232.186	347.447	484.904	646.092
4	550	1	0	4.089	25.588	71.739	140.478	232.383	347.238	484.950	646.435
5	613	1	0	4.089	25.593	71.686	140.609	232.189	347.401	485.079	645.855
6	133	1	20	4.073	25.568	71.624	140.323	231.864	346.328	483.945	644.903
7	280	1.2	20	4.073	25.555	71.445	140.116	231.936	346.097	483.238	644.674
8	410	1	20	4.080	25.549	71.579	140.300	231.744	346.819	483.831	645.103
9	570	1	20	4.082	25.512	71.586	140.209	231.700	346.824	483.324	645.633
10	962	0.6	10	4.088	25.616	71.727	140.571	232.430	347.305	485.241	646.262

**Table 4 sensors-22-01118-t004:** Calculated RFS values for the defined damage scenarios.

Scen.	Crack Pos. *x* (mm)	Crack Depth *a*_2_ (mm)	Weak Clamp. ${\bar{a}}_{1}\;(\%)$	RFS for the First 8 Modes							
				RFS 1	RFS 2	RFS 3	RFS 4	RFS 5	RFS 6	RFS 7	RFS 8
1	100	1	0	0.002393	0.001227	0.000540	0.000099	0.000015	0.000075	0.000336	0.000652
2	150	1	0	0.001899	0.000254	0.000038	0.000552	0.001173	0.001254	0.000798	0.000219
3	400	1	0	0.000644	0.001020	0.000683	0.000299	0.001480	0.000037	0.001166	0.000771
4	550	1	0	0.000240	0.001535	0.000256	0.001080	0.000632	0.000640	0.001072	0.000240
5	613	1	0	0.000141	0.001343	0.000991	0.000146	0.001467	0.000171	0.000805	0.001136
6	133	1	20	0.002224	0.000466	0.000003	0.000336	0.001015	0.001417	0.00131	0.00077
7	280	1.2	20	0.004237	0.002804	0.004348	0.003654	0.002554	0.003922	0.004598	0.002963
8	410	1	20	0.002522	0.003063	0.002483	0.002346	0.003381	0.001844	0.003377	0.002300
9	570	1	20	0.000218	0.00167	0.000507	0.000837	0.00116	0.000208	0.001627	0.00002
10	962	0.6	10	0.000412	0.000412	0.000414	0.000419	0.000429	0.000446	0.000472	0.000507

**Table 5 sensors-22-01118-t005:** Results obtained using Random forest rough model—first step.

FEM Scenarios			Random Forest Preliminary Output					
Scen.	Position (mm)	Severity *γ*_2_ (*a*_2_)	Position (mm)	Severity *γ*_2_ (*a*_2_)	Position Error (%)	Severity Error (%)	Weak Clamping	
1	100	0.0033459	99.5	0.0028	0.05	0.05	Not detected	True
2	150	0.0033459	151.6	0.0029	0.16	0.04	Not detected	True
3	400	0.0033459	458.0	0.0024	5.80	0.09	Not detected	True
4	550	0.0033459	515.8	0.0028	3.42	0.05	Not detected	True
5	613	0.0033459	602.5	0.0028	1.05	0.05	Not detected	True
6	133	0.0033459	138.1	0.0027	0.51	0.06	Not detected	False
7	280	0.0051239	204.9	0.0068	7.51	0.17	Detected	True
8	410	0.0033459	365.1	0.0044	4.49	0.11	Detected	True
9	570	0.0033459	547.3	0.0063	2.27	0.30	Detected	True
10	962	0.0011911	941.1	0.0129	2.09	1.17	Not detected	False

**Table 6 sensors-22-01118-t006:** Results obtained using Random forest enhanced model—second step.

FEM Scenarios			Refined Random Forest Output					
Scen.	Position (mm)	Severity *γ*_2_ (*a*_2_)	Position (mm)	Severity *γ*_2_ (*a*_2_)	Position Error (%)	Severity Error (%)	Weak Clamping	
1	100	0.0033459	97.7	0.0030	0.23	0.03	Detected	False
2	150	0.0033459	148.3	0.0029	0.17	0.04	Detected	False
3	400	0.0033459	400.0	0.0025	0	0.08	Not detected	True
4	550	0.0033459	539.6	0.0028	1.04	0.05	Detected	False
5	613	0.0033459	615.3	0.0026	0.23	0.07	Not detected	True
6	133	0.0033459	136.3	0.0024	0.33	0.09	Not detected	False
7	280	0.0051239	303.4	0.0087	2.34	0.36	Detected	True
8	410	0.0033459	410.1	0.0046	0.01	0.13	Detected	True
9	570	0.0033459	570.1	0.0058	0.01	0.25	Detected	True
10	962	0.0011911	976.3	0.010	1.43	0.88	Detected	True

**Table 7 sensors-22-01118-t007:** Results obtained using *network 1*—coarse estimation.

FEM Scenarios			Coarse Network—Network 1 Output					
Scen.	Position (mm)	Severity *γ*_2_ (*a*_2_)	Position (mm)	Severity *γ*_2_ (*a*_2_)	Position Error (%)	Severity Error (%)	Weak Clamping	
1	100	0.0033459	99.6	0.0027	0.04	0.06	Not detected	False
2	150	0.0033459	147.7	0.0023	0.23	0.10	Not detected	False
3	400	0.0033459	392.9	0.0011	0.71	0.22	Not detected	False
4	550	0.0033459	553.9	0.0031	0.39	0.02	Not detected	False
5	613	0.0033459	604.4	0.0022	0.86	0.11	Not detected	False
6	133	0.0033459	132.9	0.0047	0.01	0.14	Detected	True
7	280	0.0051239	261.5	0.0098	1.85	0.47	Detected	True
8	410	0.0033459	377.5	0.0033	3.25	0.00	Detected	True
9	570	0.0033459	569.1	0.0072	0.09	0.39	Detected	True
10	962	0.0011911	971	0.0134	0.9	1.22	Detected	True

**Table 8 sensors-22-01118-t008:** Results obtained using the particular networks—fine estimation.

FEM Scenarios			Accuracy Enhanced Network Output						
Scen.	Position (mm)	Severity *γ*_2_ (*a*_2_)	Network Used	Position (mm)	Severity *γ*_2_ (*a*_2_)	Position Error (%)	Severity Error (%)	Weak Clamping	
1	100	0.0033459	Sector 1	99.6	0.003332	0.04	0.00	Not detected	True
2	150	0.0033459	Sector 2	148.9	0.003174	0.11	0.02	Not detected	True
3	400	0.0033459	Sector 4	398.3	0.003031	0.17	0.03	Not detected	True
4	550	0.0033459	Sector 5	554.7	0.002947	0.47	0.04	Not detected	False
5	613	0.0033459	Sector 6	612.6	0.002936	0.04	0.04	Not detected	True
6	133	0.0033459	Sector 1	132.3	0.003385	0.07	0.00	Detected	True
7	280	0.0051239	Sector 3	280.8	0.004469	0.08	0.07	Detected	True
8	410	0.0033459	Sector 4	409.4	0.003086	0.06	0.03	Detected	True
9	570	0.0033459	Sector 6	570.4	0.003353	0.04	0.00	Detected	True
10	962	0.0011911	Sector 9	962.2	0.001419	0.02	0.02	Detected	True

**Table 9 sensors-22-01118-t009:** Frequencies estimated for the undamaged beams.

Test Beam	Natural Frequencies [Hz]							
	Mode 1	Mode 2	Mode 3	Mode 4	Mode 5	Mode 6	Mode 7	Mode 8
Beam 1	4035	25,284	70,970	139,090	230,336	344,196	481,809	641,261
Beam 2	4060	25,439	71,426	139,902	231,038	344,750	482,503	641,823
Beam 3	4034	25,341	71,064	13,915	230,138	341,868	480,773	636,769
Beam 4	4030	25,367	71,213	139,342	230,295	343,254	480,795	639,510
Beam 5	4044	25,482	71,287	139,42	228,528	344,177	481,213	641,114

**Table 10 sensors-22-01118-t010:** Estimated natural frequencies for the test beams containing a crack of known position and depth.

Test Beam	Crack Position	Crack Depth	Natural Frequencies [Hz]							
			Mode 1	Mode 2	Mode 3	Mode 4	Mode 5	Mode 6	Mode 7	Mode 8
1	98	2.5	3.952	25.086	70.854	139.086	229.859	342.143	477.105	633.647
2	310	1.25	4.053	25.422	71.259	139.818	230.913	343.835	481.913	641.729
3	569	2.5	4.024	24.904	70.707	137.883	227.494	340.994	472.791	636.758
4	126	2.5	4.200	25.226	71.208	138.982	228.245	338.441	473.771	632.801
5	759	2.5	4.043	25.343	70.051	137.012	227.517	343.834	475.326	630.614

**Table 11 sensors-22-01118-t011:** Calculated RFS value for each test beam.

Test Beam	Crack Position	Crack Depth	Calculated RFS Values							
			Mode 1	Mode 2	Mode 3	Mode 4	Mode 5	Mode 6	Mode 7	Mode 8
1	98	2.5	0.020610	0.007828	0.001629	0.000031	0.002070	0.005964	0.009762	0.011873
2	310	1.25	0.001795	0.000660	0.002334	0.000600	0.000542	0.002654	0.001223	0.000146
3	569	2.5	0.002382	0.017252	0.005019	0.009109	0.011488	0.002556	0.016603	0.000017
4	126	2.5	0.023458	0.005550	0.000064	0.002581	0.008901	0.014021	0.014610	0.010491
5	759	2.5	0.000288	0.005461	0.017336	0.017272	0.004422	0.000996	0.012234	0.016377

**Table 12 sensors-22-01118-t012:** Results obtained using the preliminary Random Forest model.

Damage Scenarios			Preliminary Random Forest					
Scen.	Position (mm)	Severity *γ*_2_ (*a*_2_)	Position (mm)	Severity *γ*_2_ (*a*_2_)	Position Error (%)	Severity Error (%)	Weak Clamping	
1	98	0.026224	96.99	0.0264	0.10	0.02	Not detected	True
2	310	0.005124	309.85	0.0051	0.01	0.00	Not detected	True
3	569	0.026224	564.87	0.0331	0.41	0.69	Not detected	True
4	126	0.026224	125.42	0.0329	0.06	0.67	Not detected	True
5	759	0.026224	758.25	0.0329	0.08	0.67	Not detected	True

**Table 13 sensors-22-01118-t013:** Results obtained using the ANN—Network 1.

Damage Scenarios			Coarse ANN—Network 1 Output					
Scen.	Position (mm)	Severity *γ*_2_ (*a*_2_)	Position (mm)	Severity *γ*_2_ (*a*_2_)	Position Error (%)	Severity Error (%)	Weak Clamping	
1	98	0.026224	97.8	0.0282	0.02	0.20	Not detected	False
2	310	0.005124	313.2	0.0042	0.32	0.09	Not detected	False
3	569	0.026224	567.2	0.0323	0.18	0.61	Not detected	False
4	126	0.026224	126	0.0337	0.00	0.75	Not detected	False
5	759	0.026224	757.8	0.0331	0.12	0.69	Not detected	False

**Table 14 sensors-22-01118-t014:** Obtained results using the refined Random Forest model.

Damage Scenarios			Refined Random Forest					
Scen.	Position (mm)	Severity *γ*_2_ (*a*_2_)	Position (mm)	Severity *γ*_2_ (*a*_2_)	Position Error (%)	Severity Error (%)	Weak Clamping	
1	98	0.026224	99.40	0.0260	0.14	0.02	Not detected	True
2	310	0.005124	309.73	0.0049	0.03	0.02	Not detected	True
3	569	0.026224	568.22	0.0327	0.08	0.65	Not detected	True
4	126	0.026224	127.84	0.0329	0.18	0.67	Not detected	True
5	759	0.026224	758.19	0.0328	0.08	0.66	Not detected	True

**Table 15 sensors-22-01118-t015:** Obtained results using the precision sector ANN.

Damage Scenarios			Enhanced Network Output					
Scen.	Position (mm)	Severity *γ*_2_ (*a*_2_)	Position (mm)	Severity *γ*_2_ (*a*_2_)	Position Error (%)	Severity Error (%)	Weak Clamping	
1	98	0.026224	98	0.0275	0.00	0.13	Not detected	True
2	310	0.005124	310	0.0054	0.00	0.03	Not detected	True
3	569	0.026224	568	0.0343	0.10	0.81	Not detected	True
4	126	0.026224	126	0.0343	0.00	0.81	Not detected	True
5	759	0.026224	758	0.0343	0.10	0.81	Not detected	True

**Table 16 sensors-22-01118-t016:** Estimated natural frequencies.

Test Beam	Crack Position	Crack Depth	Calculated Natural Frequencies [Hz]							
			Mode 1	Mode 2	Mode 3	Mode 4	Mode 5	Mode 6	Mode 7	Mode 8
Undamaged with ideal clamping			4035	25,284	70,970	139,090	230,336	344,196	481,809	641,261
Undamaged with non-ideal clamping			4.0051	25.111	70.48	138.95	229.21	341.18	476.93	635.49
1	98	2.5	3.926	24.935	70.420	138.211	228.362	339.873	474.057	630.080

**Table 17 sensors-22-01118-t017:** Calculated RFS values for the improperly clamped damaged beam.

Test Beam	Crack Position	Crack Depth	Calculated RFS Values							
			Mode 1	Mode 2	Mode 3	Mode 4	Mode 5	Mode 6	Mode 7	Mode 8
1i	98	2.5	0.026955	0.013803	0.007749	0.006322	0.008572	0.012559	0.016089	0.017436
1n-i	98	2.5	0.019749	0.007008	0.000851	0.005318	0.003699	0.003830	0.006023	0.008513

**Table 18 sensors-22-01118-t018:** Crack assessment for a damaged beam with non-ideal clamping for the case the undamaged beam had initially an ideal fixing (case 1i).

Damage Scenario			Results Obtained with the ML Models					
Scen.	Position (mm)	Severity *γ*_2_ (*a*_2_)	Position (mm)	Severity *γ*_2_ (*a*_2_)	Position Error (%)	Severity Error (%)	Weak Clamping	
First-step RF	98	0.026224	129.36	0.038300	3.14	1.21	Detected	True
Second-step RF			92.68	0.039000	0.53	1.28	Detected	True
First-step ANN			102.07	0.031283	0.41	0.51	Detected	True
Second-step ANN			99.13	0.027038	0.11	0.08	Detected	True

**Table 19 sensors-22-01118-t019:** Crack assessment for a damaged beam with non-ideal (but unchanged) clamping (case 1ii).

Damage Scenarios			Results Obtained with the ML Models					
Scen.	Position (mm)	Severity *γ*_2_ (*a*_2_)	Position (mm)	Severity *γ*_2_ (*a*_2_)	Position Error (%)	Severity Error (%)	Weak Clamping	
First-step RF	98	0.026224	82.54	0.0353	1.546	−0.9076	Not detected	True
Second-step RF			32.05	0.0292	6.595	−0.2976	Not detected	True
First-step ANN			74.63	0.0247	2.337	0.1524	Not detected	False
Second-step ANN			96.4	0.0247	0.16	0.1524	Not detected	True

## Data Availability

The data presented in this study are openly available in Mendeley at DOI:10.17632/db94d5ccr6.1 and DOI:10.17632/dn4pxx6b3m.1.

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
