# Peer review of "Beam Damage Assessment Using Natural Frequency Shift and Machine Learning"

_sensors, 2022, doi:10.3390/s22031118_

Round 1

Reviewer 1 Report

In my opinion, the presented work is well prepared, the text of the manuscript is written in an understandable manner. The applied method, as well as the presented results of experimental verification, seems to be correct. 

However, from the practical point of view the analyzed structure, namely simple steel beam, is a relatively simple structure. In the case of much more complicated objects, the creation of the database containing information about different damage scenarios could be very difficult or even impossible. Therefore the possibility of the practical application of the presented method could be discussed shortly in the text.

I recommend this manuscript for publication in the Sensors journal.     

Author Response

Dear reviewer,

thank you for the careful check of the paper and for the competent comments you made to guide us to improve it. We followed your recommendations and modified the paper in order to properly respond your questions.

The response is in the attached file. The changes in the revised version are highlighted with yellow.

Sincerly

GR Gillich

Reviewer 2 Report

The authors provide a damage assessment method based on natural frequency shift and machine learning.The random forest and artificial neural network were used as the search tool to estimate the crack location and severity. Some numerical results are shown to prove the reliability of the presented identification approach.Nonetheless,a few points should be addressed before I can recommend publication.

  • Please explain the reason for using the equation 1?Is this mathematical relationship specific to the proposed method?
  • In equation 3, the deformation information should be determined in advance.However,it is worth discussing to choose a suitable measurement method?
  • According to Figure 1 and Figure 2,the damage has little effect on RFS.Please explain how to distinguish the effects of system disturbances.
  • The RM requires a large number of measurement data , which is difficult to obtain in actual engineering.
  • Please clarify the novelty of your approach as compared to the other Structural health monitoring. This is not really clear to me.

Author Response

(The authors gave the same response as above.)

Reviewer 3 Report

In this paper, the authors proposed two machine learning methods, namely the random forest (RF) and the artificial neural network (ANN) as search tools to detect beam damage. The databases they developed contain damage scenarios for a prismatic cantilever beam with one crack and ideal and non-ideal boundary conditions. It is a carefully done study and the findings are of considerable interest. However, I thought it still has some deficiencies and I recommend to a major revision before acceptable publication. Detailed comments are listed below:

--Section 1: A more detailed description of previous research in introduction is needed, especially the published literatures on the machine learning methods in other fields, for example: Discrimination of mining microseismic events and blasts using convolutional neural networks and original waveform, Journal of Central South University, 2020, 27(10): 3078-3089. Application of signal processing and support vector machine to transverse cracking detection in asphalt pavement, Journal of Central South University, 2021, 28(8): 2451-2462. The state-of-the-art review on applications of intrusive sensing, image processing techniques, and machine learning methods in pavement monitoring and analysis. Engineering, 2021, 7(6): 845-856.
--Section 1: This part is aimed at the introduction of damage detection, however, the author only introduced the research progress of the previous author, and did not summarize and analyze it, and the literature review should be concise and try to draw general conclusions. (Lines 46- Lines 62)
--Section 1: In the introduction to the artificial neural network (ANN), the author did not give a detailed introduction to its application in damage monitoring.
--Section 2: In Figure 1 we represent the RFS calculated using equation (2) for vibration modes 1 and 3, for two crack depths. How are the parameters of these two crack depths set? How to interpret the relative frequency shift of the two crack depths in Figure 1? Please make it clearly.
--Section 2: How were Equations 1 to 6 obtained, and are there any references? Please make it clearly.
--Section 2: How to ensure that the database created is reasonable and sufficient? Will there be too many or too few cases in the database? What does this mean for the selection of intelligent algorithms? Please make it clearly.
--Section 3: What are the advantages of using machine algorithms to carry out assess beam damage over image recognition? Please make it clearly.
--Section 3: What are the advantages of artificial neural networks and random forests over other machine learning algorithms? Why did the authors choose these two methods for damage assessment?
--Section 3: When using random forests for regression, predictions beyond the training set data cannot be made, which can lead to overfitting when modeling certain noisy data. How can the author avoid this drawback?
--Table 2: What is the setting principle of random forest hyper parameters? Please make it clearly.
--Section 3: The author first introduced the ANN in section 1, and then introduced the RF. It is recommended to follow this order in section 3.
--Figure 6: Some of the letters in Figure 6 are not clear, please replace the picture with a higher resolution
--Section 4: How the parameters of the numerical simulation method are set? Please make it clearly.

Author Response

(The authors gave the same response as above.)

Reviewer 4 Report

The authors presented RF and ANN for beam damage assessment using natural frequency shift and machine learning. The paper is well-articulated and fairly presented. The following changes should be made before the paper should be considered for publication in sensors:

  1. The authors need to provide a holistic explanation of why RF and ANN are used when there are other prominent methods available for similar analysis.
  2. Although it might be general knowledge on how to tweak hyperparameters, the authors are suggested to provide a detailed explanation of how parameters are tuned. The authors are suggested to cite the following similar paper for tuning the hyperparameters.

    Sony, S. 2021. “Towards multiclass damage detection and localization 
    using limited vibration measurements.” Ph.D. thesis, Department of Civil and Environmental Engineering, University of Western Ontario, Canada.
  3. ANN is prone to overfitting, how did authors prevent overfitting. A few sentences is required to justify the method used. 
  4. The following papers should be cited in literature review/introduction to further enrich the paper

(a) Li, M., Jia, D., Wu, Z., Qiu, S., & He, W. (2022). Structural damage identification using strain mode differences by the iFEM based on the convolutional neural network (CNN). Mech. Syst. Sig. Process., 165, 108289. 

(b) Li, M., Wu, Z., Yang, H., & Huang, H. (2021). Direct damage index based on inverse finite element method for structural damage identification. Ocean Eng., 221, 108545.

(c) Al-Nasar, M. K. R., & Al-Zwainy, F. M. S. (2022). A systematic review of structural materials health monitoring system for girder-type bridges. Mater. Today:. Proc. doi: 10.1016/j.matpr.2021.12.385

Author Response

(The authors gave the same response as above.)

Round 2

Reviewer 3 Report

Accept in present form

Author Response

Dear reviewer,

thank you for the careful check of the paper and for the competent comments you made to guide us to improve it. We followed your recommendations and modified the paper in order to properly respond your questions. Also, thank you for the kind appreciation. 

Best regards

The authors

Reviewer 4 Report

The authors incorporated all the suggestions and made extensive changes. I recommend the paper for publication. 

Author Response

(The authors gave the same response as above.)
